# Beyond the mother-child dyad: Is co-residence with a grandmother associated with adolescent girls' family planning knowledge?

Emilia Zevallos-Roberts[1¤]*, Kenda Cunningham[1,2], Ramesh Prasad Adhikari[2‡], Basant Thapa[2‡], Rebecca Sear[1]

1 Department of Population Health, London School of Hygiene and Tropical Medicine, London, United Kingdom, 2 Suaahara II, Helen Keller International, Kathmandu, Nepal

These authors contributed equally to this work.

¤ Current address: Department of Research and Scholarship, Kaiser Permanente Bernard J. Tyson School of Medicine, Pasadena, California, United States of America

‡ RPA and BT also contributed equally to this work.

* Emilia.c.zevallos-roberts@kp.org

**Data Availability Statement:** The data underlying the results presented in the study are available here: https://doi.org/10.7910/DVN/4FG501 Full

## Abstract

### Background

In South Asian countries, adolescent girls are generally embedded in multigenerational households. Nevertheless, public health research continues to focus on the nuclear family and overlook the role of grandmothers in adolescent socialization and the transfer of health information. This study compares family planning knowledge of adolescent girls in households with and without a resident grandmother. Two main types of family planning knowledge were assessed: (1) modern contraceptive knowledge and (2) healthy timing and spacing of pregnancy knowledge.

### Methods

This study is a secondary data analysis of the 2017 *Suaahara* II cross-sectional survey in 16 of Nepal's 77 districts. Family planning knowledge among 769 adolescent girls was assessed and compared between those living with a grandmother (n = 330) and those not living with a grandmother (n = 439). An analysis of the relationship between co-residence and family planning knowledge was carried out using multivariate logistic regression, adjusting for potential confounders and clustering. Additionally, we used the same method to analyze the association between grandmothers' family planning knowledge and that of co-resident adolescents.

### Results

The odds of correct adolescent modern family planning knowledge were 1.81 (95% CI = 1.27,2.58) times higher in households with a grandmother. The study also identified higher odds of adolescent knowledge of modern contraceptives in households where grandmothers also had correct knowledge (OR 2.00, 95%, CI = 0.97,4.11), although this association was not statistically significant at the 0.05 alpha level. There was insufficient evidence to

citation: Zevallos-Roberts, Emilia, 2022, "Replication data for "Beyond the mother-child dyad: Is co-residence with a grandmother associated with adolescent girls' family planning knowledge?"", https://doi.org/10.7910/DVN/4FG501, Harvard Dataverse, V1, UNF:6: b4gSv7TjYV0egl/1P97RVw== [fileUNF].

**Funding:** The author(s) received no specific funding for this work.

**Competing interests:** The authors have declared that no competing interests exist.

support the association between grandmother's co-residency and correct adolescent knowledge of the healthy timing and spacing of pregnancy.

## Conclusion

This study provides support for expanding adolescent reproductive health to include the role of senior women in promoting and transmitting health care knowledge to younger women in the household.

## Introduction

Pregnancy complications are a leading cause of death among 15 to 19-year-old girls worldwide [1]. Providing reproductive health services ensures girls and women can safely manage their reproductive lives. Avoiding rapid successive pregnancies during adolescence is fundamental to maternal and infant health and the social and economic wellbeing of girls. Adolescent pregnancy in low- and middle- income countries (LMIC) has been found to be independently associated with increased risks of pre-term delivery and low birthweight babies [2]. Furthermore, pregnant adolescents drop out of school, and are less likely to participate in the labor force compared to their non-pregnant peers [1]. The health, social, and economic consequences of adolescent pregnancy perpetuate cycles of poverty to successive generations [3, 4]. Short inter-pregnancy intervals compound these effects with an independent set of health risks such as preterm births, low birthweight, and infant and early childhood mortality [5, 6].

Despite the considerable health and socioeconomic consequences of adolescent pregnancy and short pregnancy intervals, 16 million adolescents give birth every year and 95% of these births occur in low and middle-income countries (LMICs) [7]. South Asia has the second highest rate of adolescent pregnancy after Sub-Saharan Africa, and adolescent pregnancy and childbirth in Nepal remains the second highest in South Asia [4]. Nepal has attempted to tackle the underlying factors contributing to adolescent births by banning marriage under the age of 20 and legalizing abortion in 2002 [8, 9]. Despite these efforts, a 2016 nationally representative survey found that 27% of women aged 15–19 in Nepal were married compared to only 6% of men in the same age group [10]. While Nepal has been building out its adolescent-friendly health program since 2008, studies have found that only a small minority of adolescents in Nepal utilize community clinics offering sexual and reproductive health services [10].

Accurate sexual and reproductive health knowledge provides a foundation on which to build good sexual health practices [11]. Healthy Timing and Spacing of Pregnancy (HTSP) is a United States Agency for International Development (USAID)-sponsored advocacy, educational, and monitoring tool to support women to achieve their healthiest fertility in accordance with their desired family size [12]. USAID's program advocates three core principles for improved maternal and infant health: (i) women should delay their first pregnancy until the age of 20, (ii) after a live birth, women should wait 24 months before attempting their next pregnancy, and (iii) after a miscarriage or induced abortion, women should wait 6 months before attempting their next pregnancy [13]. However, public health campaigns directed at improving knowledge and attitudes may, by themselves, be insufficient to change long-run behaviors [14, 15]. Promoting optimal timing for a healthy pregnancy timing fits within broader family planning (FP) goals and must go hand in hand with health education and access to modern methods of contraception.

Knowledge of recommended pregnancy timing, desired fertility, and an adolescent's agency over their fertility do not develop in a vacuum. The United Nations Population Fund (UNFPA) suggests creating interventions that are aimed at six key stakeholders who can influence and support young women and girls: policymakers, program managers, parents, peers, partners, and providers. According to UNFPA, these stakeholders "can foster or derail adolescent development" [3]. This paper, however, seeks to understand another key, yet often overlooked, participant in adolescent development: grandmothers.

Intergenerational households are common throughout South Asia, including in Nepal. In many countries in South Asia, including Nepal, as many as 75% of the elderly report living with their children and almost 30% of households include at least one person age 60 or older [16]. Furthermore, recent transformations in family arrangements due to labor migration have accentuated the role of grandparents as caregivers [17]. Grandmothers play a pivotal role as advisors for young women in many non-western settings [18]. Despite their prominent role, public health programs continue to focus on the mother-child dyad, ignoring patterns of social organization that are especially important in collectivist cultures [19]. Martin et al. conducted a mixed-methods appraisal of behavioral interventions in LMICs that engaged fathers and grandmothers in maternal, infant, and young child nutrition. The authors discovered that most of the studies reported a positive impact of these interventions on breastfeeding rates and on family members' knowledge, attitude, and support for maternal, infant and young child nutrition [20]. Using data from Nepal's *Suaahara* program, a USAID-funded multi-sectoral nutrition initiative, Cunningham et al. conducted path analyses and concluded that a grandmother's correct knowledge of infant and young child feeding translated into a mother's correct knowledge and, ultimately, optimal infant and young child feeding practices [17].

A 2021 special issue in Maternal & Child Nutrition compiled 11 articles that contribute evidence supporting the application of family systems framework to inform nutrition programs [21]. Several studies identify grandmothers as key influencers of health knowledge within the household and conclude that child nutrition interventions might benefit by intentionally including them [21–25]. For instance, Pike et al. found that pregnant adolescents in Bangladesh reported feeling overwhelmed and inadequate to make decisions for themselves as young, first time mothers. Therefore, the adolescents identified their pregnancy as a period of increased reliance on family members for advice, guidance, and support. The study concludes that older women are "gatekeepers for health-seeking behavior" and are empowered with the culturally relevant information and knowledge to enable positive behavior change [24]. Echoing this finding, in a study of the diffusion of adolescent reproductive health knowledge in Bangladesh, the majority of girls reported that the main source of information about their periods and childbirth came from their grandmothers, older sisters, sisters-in-laws, or friends [26]. If grandmothers play such a pivotal role for women during pregnancy and postpartum, does it follow that an adolescent girls' co-residence with her grandmother would also impact her knowledge towards reproductive and sexual health concepts?

An adolescent's understanding of reproductive and sexual health is shaped by a complex ecosystem made up of individuals, institutions, and cultural value systems. In a context like Nepal, where intergenerational homes are common, a narrow, nuclear focus risks overlooking other important influencers. Bronfenbrenner's Bioecological Systems Theory supports this view, positing that adolescents are influenced by interactions with systems within and beyond their nuclear family [27]. Generally, public health interventions have focused on a primary relationship such as mother-child. Focusing on a singular relationship (child-mother) instead of the micro system in which adolescents are embedded may ignore other key mentors such as grandparents. Using Bronfenbrenner's different systems allows for analysis beyond a mother-

child or nuclear family focus to a more nuanced view of household and community drivers of sexual and reproductive health among adolescent girls.

Nepal has identified adolescents as a vulnerable and under-served population, and yet very few studies and programs have been explicitly designed to better understand and support this population [28]. The relatively scarce research on grandmothers mainly concentrates on their influence on maternal health and nutrition during pregnancy and on infant and young child nutrition. This study helps to fill a gap in understanding of the role of grandmothers by exploring how the presence or absence of a grandmother and their knowledge influenced adolescent girls' knowledge and attitudes towards modern contraceptives and healthy timing and spacing of pregnancies.

It was hypothesized that living with a grandmother, particularly a knowledgeable grandmother, would be associated with greater reproductive health knowledge of adolescent girls—more specifically, modern contraceptives and healthy timing and spacing of pregnancy.

## Methods

The paper presents a secondary analysis of a 2017 cross-sectional monitoring survey to measure the progress of *Suaahara* II, an ongoing USAID-funded multisectoral nutrition program covering all communities within 42 of Nepal's 77 districts. *Suaahara* II (2016 to 2023) aims to reduce the prevalence of maternal and child undernutrition via interventions that span health and family planning, such as nutrition, agriculture, water, sanitation and hygiene, governance, gender equality, and social inclusion.

The dataset used in this paper employed a five-stage cluster sampling design, using probability proportional to size: 1) districts (n = 16), 2) two municipalities per district(n = 32), 3) three "new" wards per municipality (n = 96), and 4) two "old" wards per new ward (n = 192), due to the large size of the "new" wards. Finally, nineteen households with a child under 5 years of age (n = 3,648) were randomly selected from a list.

Ethics approval for this project, involving secondary data analysis, was approved by the LSHTM Combined Academic, Risk Assessment, and Ethics (CARE) board (reference number: 21988, S1 Appendix). The original ethical approval for *Suaahara* II granted to Hellen Keller International by the Nepal Health Research Council, was approved on March 9[th], 2017 (reference number: 1620).

The primary survey respondents were mothers of children under age 5 and a secondary respondent was a primary household decision maker (male or female, if male unavailable). Other secondary respondents, if residing in the same household, were grandmothers of the child under age 5 and a randomly selected adolescent girl (10–19 years).

This analysis focused on unmarried adolescent girls (n = 769) and, thus, households without an adolescent girl and married adolescents were excluded from the analysis. An adolescent was identified as co-residing in the same household as a grandmother if a grandmother was present in the household during the time of the interview. See Fig 1 for a visual representation of the study population.

The primary outcome variables for this study were four measures of FP knowledge: (1) modern contraception, (2) ideal age at first pregnancy, (3) ideal spacing between birth and next pregnancy, and (4) ideal spacing between miscarriage and next pregnancy. All four variables were generated from open-ended survey questions and all responses were re-coded as binary outcomes. Modern contraceptive knowledge was categorized as either no knowledge of modern methods or knowing at least one modern method (oral contraceptive pill, condoms, injectables, intra-uterine contraceptive device (IUCD), implants, female sterilization or male sterilization. Correct knowledge of healthy timing and spacing of pregnancy was defined as

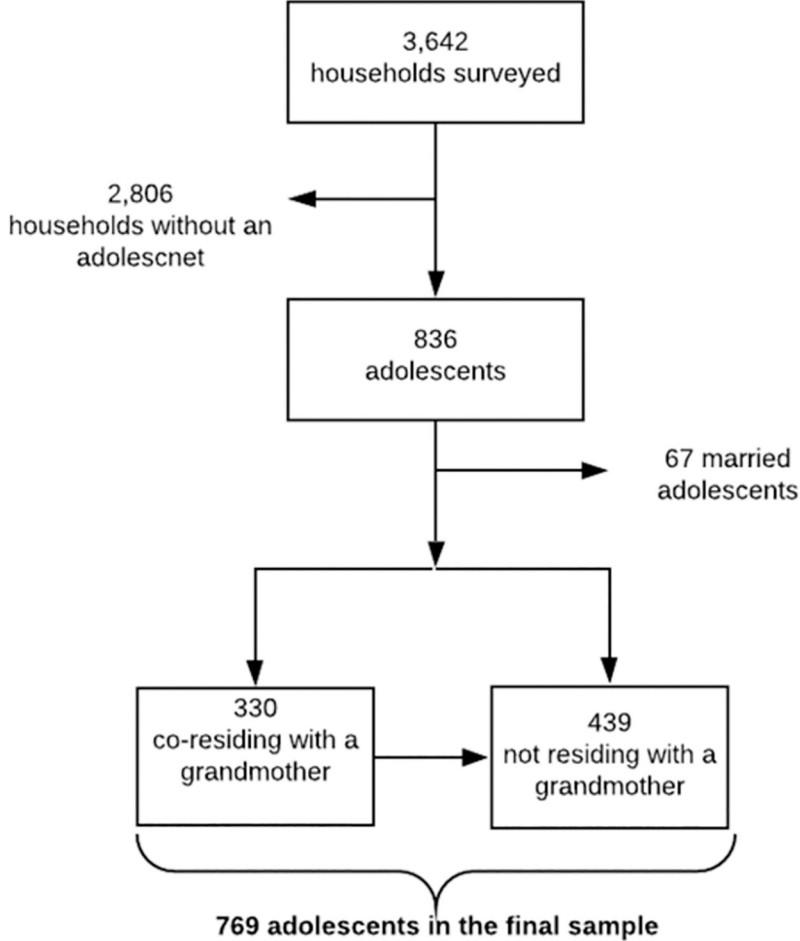

**Fig 1. Flow chart to obtain study population from original sample.**

answering with the exact correct answers: 20 years as the ideal age for first pregnancy, 24 months for ideal spacing between birth and a woman's next pregnancy, and 6 months for ideal spacing between miscarriage and a woman's next pregnancy. Analyses were also run where 'correct knowledge' was defined as a range rather than a precise figure (e.g., age at first pregnancy ≥ 20 vs. = 20), but substantive conclusions were the same.

Bronfenbrenner's Bioecological Systems Theory [27] was adapted as a framework for this study using available information collected in the survey, awareness of the local context, and findings from prior studies regarding contraception knowledge and use in Nepal and South Asia [17, 29–32]. Per Bronfenbrenner's theory, variables were grouped into their appropriate system: individual, micro-, exo- and macro- system.

Independent variables were grouped into adolescent specific, micro-, and macro-/exo- variables. Adolescent age in years was included as a variable as well as whether the individual fell into the younger (10–14) or older (15–19) age group. Adolescents also reported whether they were currently enrolled in school which was used in this analysis as an indicator of recent exposure to classroom peers and educators. Similarly, information on mass media exposure was collected by asking adolescents if and how often they watched TV, listened to the radio,

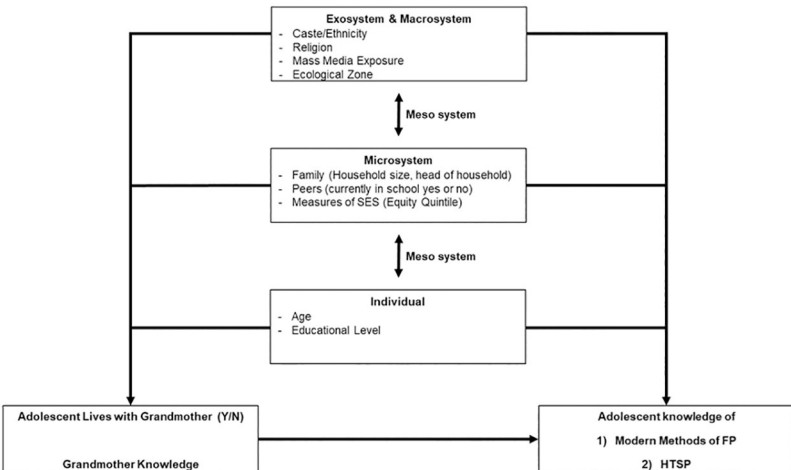

**Fig 2. Conceptual framework based on Bronfenbrenner's Ecological Systems Theory.**

read the newspaper, and used the internet. Frequent exposure to mass media was defined as having watched, listened, read, or used at least one medium once or more a week.

Differences in adolescent everyday environments—household and school—are captured through four variables at the microsystem level. Co-residence with a grandmother was counted if a grandmother of a child under age five was present in the household during the interview (so that 'grandmother co-residence' does not necessarily imply co-residence of the adolescent's own grandmother, though this is likely). Household size was collected as a continuous variable and grouped into binary categories for household size as small/average (2–5) and large (6 or more household members). Geographic location was defined by a household's agro-ecological zone and was classified into one of the three zones that divide Nepal. Each zone—*terai* [lowlands], mountain, and hill—is markedly different culturally and linguistically.

Caste/ethnicity, religion, mass media exposure, and household ecological zone were grouped into the largest and most distant systems of the adolescent's environment: exosystem and macrosystem. Equity quintiles were created based off of the results of a validated tool, EquityTool [33], that captures relative wealth based on Nepal's 2016 Demographic and Health Survey (DHS). Religion was constructed as a binary variable (Hindu and non-Hindu).

The bidirectional arrows between systems represent the mesosystem or, in other words, the interrelatedness and linkages between each level. The final framework is displayed in Fig 2. Age and education were the main individual characteristics.

## Statistical analysis

First, descriptive and bivariate analyses were carried out, followed by ANOVA and $\chi2$ tests to explore characteristics and correlations of adolescent households with (n = 330) and without (n = 409) a grandmother in co-residence. Next, two multivariate logistic regression models were built to investigate and measure the associations between (1) adolescent girl and grandmother co-residence and adolescent FP knowledge and (2) grandmother FP knowledge and adolescent FP knowledge.

All models were adjusted for adolescent age, adolescent education, household size, gender of household head, whether currently in school (as a proxy for peer interaction), household socioeconomic status, caste/ethnicity, religion, mass media exposure, and agro-ecological

zone, as well as clustering at the district level. Crude and adjusted odds ratios (ORs) and 95% confidence intervals were obtained. In the final model, variables with variance inflation factor (VIF) value larger than 5, indicating very high multicollinearity, were removed. All statistical analyses were performed using STATA 16.1.

## Results

Sociodemographic characteristics of the study population stratified by grandmother's residency status, are presented in Table 1.

Adolescents co-residing with a grandmother were significantly older and more educated: 44.9% of co-residents fell into the older age group (15–19) compared to only 24.2% of adolescents not co-residing with a grandmother. More than two-thirds (68.2%) of adolescents in co-residency with a grandmother had begun their secondary education compared to only 59.5% among adolescents not living with a grandmother.

Adolescents living with a grandmother tended to have a larger household size. Only 18.8% of adolescents living with a grandmother lived in a small/average household (2–5 members) compared to 57.2% of adolescents living without a grandmother. Furthermore, adolescents that lived with a grandmother also tended to live in male headed households: 62.4% versus 48.1%. A slightly higher proportion of households without a grandmother were classified in the lowest (29.2%) and highest (6.8%) equity quintile compared to households with a grandmother (lowest: 20.3%; highest: 3.0%). Overall, observed differences in equity quintiles were significant.

Both study populations had nearly identical religious composition and had similar distributions across agro-ecological zones. While there was no evidence to support a significant difference in household caste/ethnicity, 67% of adolescents who lived with their grandmother were exposed weekly to mass media compared to only 56% of adolescents who did not live with a grandmother.

Adolescents' FP knowledge stratified by grandmother co-residence status is presented in Table 2. Adolescents who lived with grandmothers were significantly more knowledgeable about at least one method of contraception (63.0% versus 42.4%) and across the all three contraceptive types- LARCs (23.0% versus 14.1%), SARCs (61.8% versus 40.1%), and permanent methods (18.8% versus 13.0%). Among all the adolescent girls, the best-known methods by far were SARCs, with about half of all adolescents identifying at least one type of SARC (49.4%) compared to less than a fifth of the adolescent population identifying at least one LARC (18.0%) or permanent method (15.5%).

There was no difference found in healthy timing and spacing of pregnancy knowledge between adolescents co-residing with a grandmother and adolescents not co-residing with a grandmother. Among all adolescents only 5.6% correctly knew that a woman should wait at least 6 months after a miscarriage to have another pregnancy. Similarly, very few adolescents knew that a woman should wait 24 months between giving birth and her next pregnancy (13.7%) Across the three measures of knowledge of healthy timing and spacing of pregnancy, adolescents held the greatest knowledge around the ideal age at first pregnancy with 20.3% of all adolescents correctly stating "20 years old" as the ideal age at first pregnancy. While a greater proportion of adolescents co-residing with a grandmother held correct knowledge around the ideal age at first pregnancy compared to adolescents not living with their grandmothers (22.7% versus 18.5%), the difference was statistically insignificant.

Adolescent FP knowledge stratified by grandmother knowledge is shown in Table 3. Adolescents who live with a grandmother with correct knowledge were significantly more likely to have correct knowledge of both modern contraception (65.9% versus 52.8%) and ideal age at

**Table 1. Adolescent sociodemographic characteristics stratified by grandmother residency status.**

| SAMPLE CHARACTERISTICS | TOTAL | GRANDMOTHER CO-RESIDENCE | | Test Statistic (F-statistic or $X^2$) | P |
|---|---|---|---|---|---|
| | n = 769 | NO | YES | | |
| | | n = 439 (57.1%) | n = 330 (42.9%) | | |
| | n (%) | Mean [SD], median {IQR}, or n (%) | Mean [SD], median {IQR}, or n (%) | | |
| **Individual Characteristics** | | | | | |
| Age in years (range: 10–19) | 769 (100) | 13 {3} | 14 {4} | 48.9 | <0.001* |
| Age group | | | | | |
| 10–14 | 515 (67.0) | 333 (75.9) | 182 (55.2) | 36.5 | <0.001 |
| 15–19 | 254 (33.0) | 106 (24.2) | 148 (44.9) | | |
| Education (range: 0–12) | 769 (100) | 6.0 [2.3] | 6.9 [2.4] | 28.1 | <0.001 |
| Education group | | | | | |
| None/Some Primary | 283 (36.8) | 178 (40.6) | 105 (31.8) | 6.2 | 0.013 |
| Some secondary + | 486 (63.2) | 261 (59.5) | 225 (68.2) | | |
| **Microsystem** | | | | | |
| Household size (range: 2–34) | 769 (100) | 5 {2} | 7 {3} | 138.4 | <0.001* |
| Household size category | | | | | |
| Small (2–5) | 313 (40.7) | 251 (57.2) | 62 (18.8) | 115.0 | <0.001 |
| Large (6+) | 456 (59.3) | 188 (42.8) | 368 (81.2) | | |
| HH gender | | | | | |
| Male | 417 (54.2) | 211 (48.1) | 206 (62.4) | 15.7 | <0.001 |
| Female | 352 (45.8) | 228 (51.9) | 124 (37.6) | | |
| Currently in school | | | | | |
| No | 54 (7.0) | 20 (4.6) | 34 (10.3) | 9.5 | 0.002 |
| Yes | 715 (93.0) | 419 (95.4) | 296 (89.7) | | |
| Household equity quintile | | | | | |
| Lowest | 195 (25.4) | 128 (29.2) | 67 (20.3) | 15.6 | 0.004 |
| 2nd Lowest | 231 (30.0) | 124 (28.3) | 107 (32.4) | | |
| Middle | 164 (21.3) | 84 (19.1) | 80 (24.2) | | |
| 2nd Highest | 139 (18.1) | 73 (16.6) | 66 (20.0) | | |
| Highest | 40 (5.2) | 30 (6.8) | 10 (3.0) | | |
| **Exosystem and Macrosystem** | | | | | |
| Religion | | | | | |
| Hindu | 692 (90.0) | 398 (90.7) | 294 (89.1) | 0.5 | 0.473 |
| Other | 77 (10.0) | 41 (9.3) | 36 (10.9) | | |
| Agro-ecological zone | | | | | |
| Mountain | 122 (15.9) | 71 (16.2) | 51 (15.5) | 0.3 | 0.862 |
| Hill | 394 (51.2) | 227 (51.7) | 167 (50.6) | | |
| Terai | 253 (32.9) | 141 (32.1) | 112 (33.9) | | |
| Caste/Ethnicity | | | | | |
| Upper | 303 (39.4) | 161 (36.7) | 142 (43.0) | 3.2 | 0.074 |
| Other | 466 (60.6) | 278 (63.3) | 188 (57.0) | | |
| Mass media exposure group | | | | | |
| Infrequent | 302 (39.3) | 193 (44.0) | 109 (33.0) | 9.4 | 0.002 |
| Frequent | 467 (60.7) | 246 (56.0) | 221 (67.0) | | |
| Mass media exposure by type | | | | | |
| Radio | | | | | |
| Frequent | 223 (29.0) | 111 (25.3) | 112 (33.9) | 6.9 | 0.009 |
| Infrequent | 546 (71.0) | 328 (74.7) | 218 (66.1) | | |

*(Continued)*

**Table 1.** (Continued)

| SAMPLE CHARACTERISTICS | TOTAL | GRANDMOTHER CO-RESIDENCE | | Test Statistic (F-statistic or $X^2$) | P |
|---|---|---|---|---|---|
| | n = 769 | NO | YES | | |
| | | n = 439 (57.1%) | n = 330 (42.9%) | | |
| | n (%) | Mean [SD], median {IQR}, or n (%) | Mean [SD], median {IQR}, or n (%) | | |
| TV | | | | | |
| Frequent | 317 (41.2) | 173 (39.4) | 144 (43.6) | 1.4 | 0.238 |
| Infrequent | 452 (58.8) | 266 (60.6) | 186 (56.4) | | |
| Newspaper | | | | | |
| Frequent | 14 (1.8) | 6 (1.4) | 8 (2.4) | 1.2 | 0.278 |
| Infrequent | 755 (98.2) | 433 (98.6) | 322 (97.6) | | |
| Internet | | | | | |
| Frequent | 44 (5.7) | 16 (3.6) | 28 (8.5) | 8.2 | 0.004 |
| Infrequent | 725 (94.3) | 423 (96.4) | 302 (91.5) | | |

*Requires cautious interpretation since Bartlett's test for equal variances <0.05; In these cases, median reported rather than mean

first pregnancy (27.5% versus 17.8%) compared to adolescents who live with a grandmother without correct knowledge. Grandmothers' correct knowledge around the other two main FP knowledge questions (spacing between birth and miscarriage and next pregnancy) were not significantly associated with differences in adolescent knowledge.

To mitigate data sparsity issues and avoid biasing parameter estimates, some data fields were combined. The two highest equity quintiles were collapsed and combined into the second

**Table 2. Adolescent FP knowledge stratified by grandmother residency status.**

| ADOLESCENT KNOWLEDGE | TOTAL | GRANDMOTHER CO-RESIDENCE | | Test Statistic ($X^2$) | P |
|---|---|---|---|---|---|
| | n = 769 | NO | YES | | |
| | | n = 439 (57.1%) | n = 330 (42.9%) | | |
| | n (%) | n (%) | n (%) | | |
| **Contraception** | | | | | |
| Modern methods of contraception knowledge | | | | | |
| At least one | 394 (51.2) | 186 (42.4) | 208 (63.0) | 32.2 | <0.001 |
| None | 375 (48.8) | 253 (57.6) | 122 (37.0) | | |
| Knowledge by Type of Method | | | | | |
| At least one LARC | 138 (18.0) | 62 (14.1) | 76 (23.0) | 10.2 | 0.001 |
| At least one SARC | 380 (49.4) | 176 (40.1) | 204 (61.8) | 35.6 | <0.001 |
| At least one permanent | 119 (15.5) | 57 (13.0) | 62 (18.8) | 4.9 | 0.03 |
| **Healthy Timing and Spacing of Pregnancy Knowledge** | | | | | |
| Ideal age at first pregnancy | | | | | |
| = 20 years | 156 (20.3) | 81 (18.5) | 75 (22.7) | 2.1 | 0.144 |
| ≠ 20 years/unsure | 613 (79.7) | 358 (81.6) | 255 (77.3) | | |
| Ideal spacing between birth and next pregnancy | | | | | |
| = 24 months | 105 (13.7) | 64 (14.6) | 41 (12.4) | 0.7 | 0.389 |
| ≠ 24 months/unsure | 664 (86.4) | 375 (84.4) | 289 (87.6) | | |
| Ideal spacing between miscarriage and next pregnancy | | | | | |
| = 6 months | 43 (5.6) | 26 (5.9) | 17 (5.5) | 0.2 | 0.645 |
| ≠ 6 months/unsure | 726 (94.4) | 413 (94.1) | 313 (94.9) | | |

**Table 3. Adolescent FP knowledge stratified by grandmother knowledge.**

| ADOLESCENT KNOWLEDGE | TOTAL | GRANDMOTHER KNOWLEDGE | | Test Statistic (X$^2$) | P |
|---|---|---|---|---|---|
| | n = 330 | n (%) | | | |
| | n (%) | | | | |
| **Modern methods of contraception knowledge** | | At Least One | None | | |
| At least one | 208 (63.0) | 170 (65.9) | 38 (52.8) | 4.2 | 0.042 |
| None | 122 (37.0) | 88 (34.1) | 34 (47.2) | | |
| **Knowledge by Type of Method** | | At Least One | None | | |
| At least one LARC | 76 (23.03) | 35 (31.0) | 41 (19.0) | 6.1 | 0.013 |
| At least one SARC | 204 (61.8) | 156 (65.3) | 48 (52.8) | 4.4 | 0.036 |
| At least one permanent | 62 (18.8) | 26 (19.1) | 36 (18.6) | 0.9 | 0.017 |
| **Ideal age at first pregnancy** | | = 20 years | ≠ 20 years/unsure | | |
| = 20 years | 75 (22.7) | 46 (27.5) | 29 (17.8) | 4.5 | 0.035 |
| ≠ 20 years/unsure | 255 (77.3) | 121 (72.5) | 134 (82.2) | | |
| **Ideal spacing between birth and next pregnancy** | | = 24 months | ≠ 24 months/unsure | | |
| = 24 months | 41 (12.4) | 4 (12.9) | 37 (12.4) | 0.01 | 0.932 |
| ≠ 24 months/unsure | 289 (87.6) | 27 (87.1) | 262 (87.6) | | |
| **Ideal spacing between miscarriage and next pregnancy** | | = 6 months | ≠ 6 months/unsure | | |
| = 6 months | 17 (5.2) | 3 (10.0) | 14 (4.7) | 1.6 | 0.208 |
| ≠ 6 months/unsure | 313 (94.9) | 27 (90.0) | 26 (95.3) | | |

highest equity quintiles since only 10 (3.0%) adolescents pertained to the highest equity quintile. Furthermore, the religion variable was dropped as 98.8% of the sample identified as Hindu, and, instead, identifiers of caste and ethnicity were included in the analysis. Since few grandmothers had knowledge around the ideal spacing of pregnancies between a miscarriage and subsequent pregnancy (n = 30) and birth and subsequent pregnancy (n = 31), these two FP questions were omitted as independent variables in regression analysis investigating the effect of grandmother knowledge on adolescent knowledge. To detect multicollinearity, a VIF diagnostic test was run on each model. In light of the diagnostic results, currently enrollment in school (VIF>7) was omitted to reduce multicollinearity. Across all final models, no variable had a VIF greater than 5, suggesting non-significant collinearity among explanatory variables.

## Association between grandmother and adolescent co-residency and adolescent family planning knowledge

Table 4 presents results from the multivariate logistic regression investigating the association of grandmothers' co-residency with the adolescent girl and adolescent FP knowledge. After controlling for socio-economic factors, adolescents co-residing with grandmothers had increased odds of knowledge of at least one modern method of contraceptive 1.81 (95% CI [1.27–2.58], p = 0.001) compared to adolescents not living with a grandmother. There was no evidence of an association between adolescent knowledge of pregnancy timing and spacing and grandmother residency status.

Thirty households included grandmothers in the household roster even though the grandmothers were not present at the time of the interview. A sensitivity analysis was run to determine the impact of treating these households as not having a resident grandmother. Excluding these households from the sample reduced the number of "no grandmother" households by 30 but had very small impacts on the estimated coefficients of the primary explanatory variables in the models.

**Table 4. Crude and multivariate associations of grandmothers' residency and adolescent FP knowledge (n = 769).**

| | | CRUDE Ψ | | ADJUSTED | |
|---|---|---|---|---|---|
| | | OR (95% CI) | p† | OR (95% CI) | p† |
| **MODERN METHODS OF CONTRACEPTION** | Co-Residence (ref. no) | 2.32 (1.71–3.15) | <0.001 | 1.81 (1.27–2.58) | 0.001 |
| | Age (ref. younger) | | | 6.24 (3.98–9.79) | <0.001 |
| | Education (ref. none/primary) | | | 3.44 (2.37–5.10) | <0.001 |
| | Equity Quintile (ref. lowest) | | | | |
| | 2nd lowest | | | 1.63 (1.02–2.59) | 0.129 |
| | Middle | | | 1.45 (0.86–2.45) | |
| | Highest | | | 1.08 (0.58–2.02) | |
| | Household size (ref. small) | | | 0.98 (0.65–1.47) | 0.899 |
| | HH gender (ref. male) | | | 1.27 (0.86–1.87) | 0.255 |
| | Caste/Ethnicity (ref. other) | | | 0.90 (0.61–1.33) | 0.594 |
| | Mass Media (ref. infrequent) | | | 1.77 (1.19–2.65) | 0.005 |
| | Ecological Zone (ref. hill) | | | | |
| | Mountain | | | 1.06 0.66–1.70) | 0.681 |
| | *Terai* | | | 0.81 (0.48–1.49) | |
| **IDEAL AGE AT FIRST PREGNANCY** | Co-Residence (ref. no) | 1.30 (0.91–1.85) | 0.143 | 1.26 (0.83–1.92) | 0.282 |
| | Age (ref. younger) | | | 1.53 (0.98–2.39) | 0.01 |
| | Education (ref. none/primary) | | | 1.34 (0.86–2.39 | 0.060 |
| | Equity Quintile (ref. lowest) | | | | 0.001 |
| | 2nd lowest | | | 0.91 (5.78–1.42) | |
| | Middle | | | 0.47 (0.25–0.87) | |
| | Highest | | | 0.30 (0.16–0.56) | |
| | Household size (ref. small) | | | 0.89 (0.62–1.40) | 0.740 |
| | HH gender (ref. male) | | | 0.93 (0.62–1.42) | 0.769 |
| | Caste/Ethnicity (ref. other) | | | 0.94 (0.62–1.42) | 0.769 |
| | Mass Media (ref. infrequent) | | | 0.94 (0.65–1.38) | 0.773 |
| | Ecological Zone (ref. hill) | | | | |
| | Mountain | | | 1.17 (0.72–1.88) | 0.040 |
| | *Terai* | | | 1.91 (1.16–3.13) | |
| **IDEAL SPACING BETWEEN BIRTH AND NEXT PREGNANCY** | Co-Residence (ref. no) | 0.83 (0.55–1.25) | 0.377 | 0.83 (0.52–1.32) | 0.431 |
| | Age (ref. younger) | | | 0.61 (0.35–1.08) | 0.091 |
| | Education (ref. none/primary) | | | 1.30 90.75–2.24) | 0.344 |
| | Equity Quintile (ref. lowest) | | | | |
| | 2nd lowest | | | 1.21 (0.70–2.10) | 0.680 |
| | Middle | | | 0.88 (0.43–1.81) | |
| | Highest | | | 1.22 (0.55–2.73) | |
| | Household size (ref. small) | | | 1.16 (0.71–1.89) | 0.544 |
| | HH gender (ref. male) | | | 1.04 (0.66–1.64) | 0.865 |
| | Caste/Ethnicity (ref. other) | | | | |
| | Mass Media (ref. infrequent) | | | 1.29 (0.79–2.10) | 0.307 |
| | Ecological Zone (ref. hill) | | | | |
| | Mountain | | | 1.69 (1.05–2.72) | 0.018 |
| | *Terai* | | | 0.66 (0.36–1.20) | |

(*Continued*)

**Table 4.** (Continued)

| | | CRUDE Ψ | | ADJUSTED | |
|---|---|---|---|---|---|
| | | OR (95% CI) | p† | OR (95% CI) | p† |
| IDEAL SPACING BETWEEN MISCARRIAGE AND NEXT PREGNANCY | Co-Residence (ref. no) | 0.86 (0.47–1.60) | 0.64 | 0.68 (0.29–1.59) | 0.370 |
| | Age (ref. younger) | | | 2.03 (0.97–4.26) | 0.01 |
| | Education (ref. none/primary) | | | 2.02 (0.78–5.25) | |
| | Equity Quintile (ref. lowest) | | | | |
| | 2nd lowest | | | 1.25 (0.51–3.05) | 0.339 |
| | Middle | | | 2.22 (0.84–5.86) | |
| | Highest | | | 1.08 (0.29–3.95) | |
| | Household size (ref. small) | | | 0.75 (0.34–1.66) | 0.471 |
| | HH gender (ref. male) | | | 0.55 (0.28–1.07) | 0.079 |
| | Caste/Ethnicity (ref. other) | | | 0.73 (0.26–1.47) | 0.374 |
| | Mass Media (ref. infrequent) | | | 1.52 (0.65–3.55) | 0.330 |
| | Ecological Zone (ref. hill) | | | | |
| | Mountain | | | 2.02 (11.1–3.80) | 0.003 |
| | *Terai* | | | 0.34 (0.13–0.94) | |

Ψ Adjusted for district level clustering

† Wald test

## Association between grandmother family planning knowledge and adolescent family planning knowledge

Table 5 presents the multivariate logistic regression results investigating the association of grandmother's FP knowledge and adolescent FP knowledge among the subset of adolescents co-residing with a grandmother (n = 330). In the crude analyses, adolescent girls were more likely to know at least one method of modern contraceptive and state that the ideal age at first pregnancy (age 20) if the grandmother in the household also held correct FP knowledge. The association does not imply causality as grandmothers may be informed by adolescents who learn FP information in school, for example. In the final model adjusted for individual, micro- and macro-/exo- system factors, the odds ratios similarly suggested a positive association between grandmothers' and adolescents' knowledge of both modern contraception and ideal age at first pregnancy, but these associations were not significant at the p<0.05 level.

## Discussion

This study explored relationships between grandmother and adolescent girls' co-residency and family planning and healthy timing and spacing of pregnancy knowledge, as well as family planning knowledge among resident grandmothers and adolescent girls. The intention here was to draw attention to the role of grandmothers as advisors to adolescent women, particularly surrounding their reproductive health. Most research on grandmothers mainly concentrates on their influence on maternal, infant and child health and nutrition, although there is a growing body of literature exploring the role of extended family members in transferring reproductive and sexual health knowledge to adolescents.

After controlling for individual, household, and community factors, we found evidence of a positive association between co-residence with grandmothers and adolescent modern contraceptive knowledge. Supporting this finding are studies that explored inter-familial

**Table 5. Crude and multivariate associations between co-resident grandmothers' FP knowledge and adolescent FP knowledge (n = 330).**

| ADOLESCENT FP KNOWLEDGE | | CRUDE<sup>Ψ</sup> | | ADJUSTED | |
|---|---|---|---|---|---|
| | | OR (95% CI) | p† | OR (95% CI) | p† |
| **MODERN METHODS OF CONTRACEPTION** | Grandmother's Knowledge (ref. None) | 1.73 (1.00–2.97) | 0.048 | 2.00 (0.97–4.11) | 0.060 |
| | Age (ref. younger) | | | 7.81 (4.19–14.58) | <0.001 |
| | Education (ref. none/primary) | | | 3.34 (1.81–6.14) | <0.001 |
| | Equity Quintile (ref. lowest) | | | | |
| | 2<sup>nd</sup> lowest | | | 3.84 (1.54–9.56) | 0.016 |
| | Middle | | | 1.56 (0.62–3.90) | |
| | Highest | | | 1.30 (0.44–3.88) | |
| | Household size (ref. small) | | | 0.71 (0.33–1.53) | 0.378 |
| | HH gender (ref. male) | | | 0.92 (0.51–1.64) | 0.766 |
| | Caste/Ethnicity (ref. other) | | | 1.04 (0.54–2.00) | 0.916 |
| | Mass Media (ref. little/none) | | | 1.78 (0.88–3.60) | 0.105 |
| | Ecological Zone (ref. hill) | | | | |
| | Mountain | | | 1.07 (0.46–2.51) | 0.904 |
| | *Terai* | | | 0.84 (0.34–2.08) | |
| **IDEAL AGE AT FIRST PREGNANCY** | Grandmother's Knowledge | 1.76 (1.03–3.01) | 0.040 | 1.63 (0.92–2.91) | 0.096 |
| | Age (ref. younger) | | | 1.43 (0.90–2.56) | 0.220 |
| | Education (ref. none/primary) | | | 1.34 (0.86–2.39 | 0.373 |
| | Equity Quintile (ref. lowest) | | | | 0.078 |
| | 2<sup>nd</sup> lowest | | | 0.91 (5.78–1.42) | |
| | Middle | | | 0.47 (0.25–0.87) | |
| | Highest | | | 0.30 (0.16–0.56) | |
| | Household size (ref. small) | | | 0.55 (0.26–1.13) | 0.103 |
| | HH gender (ref. male) | | | 1.07 (0.58–1.98) | 0.831 |
| | Caste/Ethnicity (ref. other) | | | 0.69 (0.36–1.32) | 0.260 |
| | Mass Media (ref. little/none) | | | 0.67 (0.39–1.16) | 0.155 |
| | Ecological Zone (ref. hill) | | | | |
| | Mountain | | | 0.13 (0.60–2.85) | 0.379 |
| | *Terai* | | | 1.62 (0.80–3.27) | |

<sup>Ψ</sup>Adjusted for district level clustering

† Wald test

communication and transfer of sexual and reproductive health knowledge in LMICs [34–36]. A study in Nigeria focused on factors that impacted the effectiveness of reproductive knowledge transfer from mother to daughter with private, in home, and informal communication found to be most effective. The study authors recommend targeted family life education to improve reproductive health knowledge of parents as well as girls to positively impact adolescent health [34]. An impact assessment of "The Grandmother Project", part of a USAID funded "Passages Project" to improve adolescent reproductive and sexual health, found that improving grandmothers' knowledge of spacing and timing of pregnancy resulted in senior women having a better understanding of the risk of early marriage and pregnancy for adolescents. In addition, grandmothers who participated in the program felt more empowered to advocate in girls' interest with respect to timing of marriage [37].

Our study also found that the most common type of family planning knowledge among the study population of both adolescents and resident grandmothers was modern contraceptive knowledge, although the prevalence of this knowledge (72% for grandmothers, 78% for

adolescents) was much lower compared to other national estimates. The 2016 DHS, for instance, reported that 99.7% of all never-married women age 15–49 knew at least one modern method [38]. Differences in the study population and the 2016 DHS statistics are likely due to the differences in assessing knowledge (DHS uses a probed "ever heard" method rather than unaided recall) and the DHS respondents were older on average and with a broader age range. In terms of Healthy Timing and Spacing of Pregnancy, neither adolescents nor grandmothers in this survey were very knowledgeable, particularly regarding the healthy spacing between births and miscarriages and a woman's next pregnancy, as summarized in Table 3. This finding also is supported by the report on "The Grandmother Project" where, prior to the policy intervention, both grandmothers and adolescents had low knowledge of best practices for timing and spacing of pregnancy [37]. While knowledge appears to be positively associated with co-residency for both adolescents and grandmothers, the direction of knowledge transfer was unclear. However, the impact analysis of the "The Grandmother Project" suggests that adolescent girls may benefit where senior women are included in health initiatives that target this population.

## Strengths and limitations

A major strength of the present study was the quality of the data set, sampling, and data collection methods, which provided a complete data set for the variables of interest (no missing data), a widespread geographic representation of Nepal, low drop-out rate among respondents and a rare opportunity to investigate multiple generations within the same household. The data set, however, presented certain limitations. First, the cross-sectional nature of the data set inhibited determining causal relationships. Second, while the analysis controlled for observed differences between adolescents who co-resided with a grandmother and those who did not (Table 1), it is possible there were additional unmeasured differences that may confound the association between co-residence and adolescents' family planning knowledge. Also, due to the nature of how grandmother residency was defined, we were not able to identify whether adolescents were living with their own grandmother or the grandmother of another young child in the household (though the influence of 'grandmothers' in the literature often refers to senior women more broadly [18]). Furthermore, knowledge of contraception, as measured in this survey and other national surveys, is not sufficiently detailed to understand the mastery of the subject. As documented in previous studies, knowledge of a method alone may be insufficient for safe and effective use [39]. Regarding healthy timing and spacing of pregnancy knowledge, attributing correct knowledge to exact answers rather than ranges of acceptable answers likely produced a far more conservative estimation of knowledge. However, the more conservative definition was chosen to better approximate true familiarity as opposed to guesses.

While this study uses systems analysis to identify the source of sexual and reproductive knowledge among the surveyed adolescents, the survey design and data do not support the analysis of the interactions and dependencies between individual and contextual systems. For example, interactions with peer groups and schools, as well as the adolescent's sexual partnerships, were beyond the scope of this study. These nested systems and their synergetic effects, however, can play an important role in adolescent development, inviting or inhibiting adolescent FP knowledge.

Lastly, due to the rarity of some events and the smaller sample sizes in sub-population analysis, parts of the study were likely underpowered. More precisely, relatively few grandmothers and adolescents held correct knowledge around the appropriate spacing between pregnancies and miscarriage and a woman's subsequent pregnancy and the sub-population analysis included less than half of the sample population.

## Conclusion

The results of this study support the rationale to further investigate the role of grandmothers in promoting and transmitting health care knowledge to younger women in the household. Additional studies are needed to better understand the relationship between the presence or absence of a grandmother and the diffusion of knowledge from grandmother to adolescent. For instance, it would be of interest to understand whether the association between co-residence and adolescent knowledge of modern contraceptives extends to practical knowledge such as side effects, method characteristics and effectiveness and how and where to obtain contraception. Furthermore, leveraging qualitative data within an ecological systems approach may provide insights into when and how knowledge diffuses from grandmother to adolescent and lend support to the role of grandmothers as health advisors. With a better understanding of the impact of grandmothers on adolescent reproductive and sexual health knowledge and practices in Nepal and other South Asian contexts, FP initiatives could leverage a grandmother's role as caregiver to encourage the next generation's use of modern contraception and healthy timing and spacing of pregnancies.

## Supporting information

**S1 Appendix.**
(DOCX)

## Acknowledgments

We want to thank USAID for their support of *Suaahara* II and Hellen Keller International for creating and managing the primary dataset. We gratefully acknowledge all those who designed, piloted and implemented the survey, especially the New ERA data collection teams and the thousands of survey respondents who gave their time and energy to the study.

In addition, we thank Rotary International (Denver, USA Chapter) for the Fellowship that funded the first author's studies at the London School of Hygiene and Tropical Medicine (LSHTM), We also thank the London School of Hygiene and Tropical Medicine (LSHTM) faculty and staff that support the MSc in Reproductive & Sexual Health Research.

## Author Contributions

**Conceptualization:** Emilia Zevallos-Roberts, Kenda Cunningham, Rebecca Sear.

**Data curation:** Emilia Zevallos-Roberts, Ramesh Prasad Adhikari.

**Formal analysis:** Emilia Zevallos-Roberts.

**Methodology:** Kenda Cunningham, Rebecca Sear.

**Supervision:** Kenda Cunningham, Rebecca Sear.

**Writing – original draft:** Emilia Zevallos-Roberts.

**Writing – review & editing:** Emilia Zevallos-Roberts, Kenda Cunningham, Basant Thapa, Rebecca Sear.

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
