## [Decision Letter · Decision Letter 0]

5 Jan 2022

PONE-D-21-35002Beyond the mother-child dyad: Is co-residence with a grandmother associated with adolescent girls’ family planning knowledge?PLOS ONE

Dear Dr. Zevallos-Roberts,

Thank you for submitting your manuscript to PLOS ONE. After careful consideration, we feel that it has merit but does not fully meet PLOS ONE’s publication criteria as it currently stands. Therefore, we invite you to submit a revised version of the manuscript that addresses the points raised during the review process.

We look forward to receiving your revised manuscript.

Kind regards,

Kannan Navaneetham, PhD

Academic Editor

PLOS ONE

a) Did participants provide their written or verbal informed consent to participate in this study?

Reviewers' comments:

Reviewer's Responses to Questions

**Comments to the Author**

1. Is the manuscript technically sound, and do the data support the conclusions?

Reviewer #1: Yes

2. Has the statistical analysis been performed appropriately and rigorously? 

Reviewer #1: Yes

3. Have the authors made all data underlying the findings in their manuscript fully available?

Reviewer #1: Yes

4. Is the manuscript presented in an intelligible fashion and written in standard English?

Reviewer #1: Yes

5. Review Comments to the Author

Reviewer #1: Thank you for the opportunity to review this timely paper. I have welcomed reading it and particularly second the argument made by the authors on the need to grow the evidence base regarding the role that grandmothers play as advisors for adolescent girls’ sexual and reproductive health (SRH), beyond maternal and child nutrition. Grandmothers have indeed been under-represented in the public health literature and this paper contributes to filling this gap, providing a dataset analysis attained by accessing a population also under-researched in the wider literature. It will thus contribute to the wider global health call to expand research by including more evidence from global south regions.

I only have some minor comments to add in terms of suggestions for revisions:

Introduction: the rationale is sound and draws on recent evidence regarding grandmothers and wider family systems.

Line 86: “Sexual and reproductive health knowledge supports good sexual health practices.” This is a firm statement and I wonder if the authors could consider providing references, particularly given critiques that question the linearity of the knowledge-attitudes-behaviour assumption, as there are approaches applied to public health which have also highlighted that knowledge alone is not enough to change practices. This I suggest would align with the cited literature and on the ecological model the authors are drawing on, which recognises that individuals live within embedded systems beyond the nuclear focus given to the dyad.

Methods:

Line 167-170: sentence is a bit difficult to follow.

Line 221: “Frequent exposure”: Please tell us more about how this criterion was defined, could it be perhaps too arbitrary for the adolescent population in this context? How widely/not are phones/devices with internet access used to set the bar at: at least one medium once/+ a week?

Results

I appreciate the clear tables describing the results and the descriptions.

Table 4 – seems to be missing a word in the title

Discussion: I very much welcome the evidence provided in this section identifying relevant, recent projects and evidence to corroborate the role grandmothers play in SRH. But I would also like to invite the authors to re-engage with the ecological model to conceptually enhance their results. The authors present a helpful figure on the application of the model but the results also identify demographic patterns on grandmother co-residence and knowledge(s), so I suggest they could consider: how could Bronfenbrenner’s insights better inform these results beyond categorizing the variables in the individual, micro, exo and macro systems?

The limitations are clearly stated as the authors are working with an existing dataset. Their arguments for future research were aligned with my own reading of the study, calling towards exploring the knowledge exchange between co-resident grandmothers and adolescents: what types of knowledge are shared that could help better inform grandmothers’ role as advisors and influencers of FP and SRH more generally? A call that I also second and invite the authors to consider for their next research project.

6. PLOS authors have the option to publish the peer review history of their article (what does this mean?). If published, this will include your full peer review and any attached files.

Reviewer #1: No

---

## [Author Response · Author response to Decision Letter 0]

22 Feb 2022

Dear Academic Editor and Reviewer:

Thank you for your careful review and constructive comments and suggestions to the draft. Please see below a point-by-point response to these comments.

Comment: 

Line 85-86: “Sexual and reproductive health knowledge supports good sexual health practices.” This is a firm statement and I wonder if the authors could consider providing references, particularly given critiques that question the linearity of the knowledge-attitudes-behaviour assumption, as there are approaches applied to public health which have also highlighted that knowledge alone is not enough to change practices. This I suggest would align with the cited literature and on the ecological model the authors are drawing on, which recognises that individuals live within embedded systems beyond the nuclear focus given to the dyad.

Response: 

Agreed that we should be careful to note that knowledge alone is not necessarily enough to change practices. We have added a reference to WHO publication that identifies education as one of several critical elements to improving sexual health practices. We added reference to an analysis of sexual health campaigns directed at young adults and adolescents that found evidence that that public health campaigns had measurable impact on knowledge, but there is inconsistent evidence on the impact on practices. 

Sentences were revised as follows:

Line 85-86: Accurate sexual and reproductive health knowledge provides a foundation on which to build good sexual health practices [11].

11. WHO. Sexual health [Internet]. [cited 2022 Jan 18]. Available from: https://www.who.int/health-topics/sexual-health#tab=tab_1

Line 94-96: However, public health campaigns directed at improving knowledge and attitudes may, by themselves, be insufficient to change long-run behaviors [14,15]

14. Boonstra HD. Advancing Sexuality Education in Developing Countries: Evidence and Implications. Guttmacher Policy Rev. 2011;14(3):17–23. 

15. Oringanje C, Meremikwu MM, Eko H, Esu E, Meremikwu A, Ehiri JE. Interventions for preventing unintended pregnancies among adolescents. Cochrane Database Syst Rev [Internet]. 2016 Feb 3 [cited 2022 Jan 18];2016(2). Available from: https://www.cochranelibrary.com/cdsr/doi/10.1002/14651858.CD005215.pub3/full

Comment:

Methods:

Line 167-170: sentence is a bit difficult to follow.

Response:

Sentence was revised as follows (Line 169-172): Suaahara II (2016 to 2023) aims to reduce the prevalence of maternal and child undernutrition via interventions that span health and family planning, such as nutrition, agriculture, water, sanitation and hygiene, governance, gender equality, and social inclusion.

Comment:

Line 221: “Frequent exposure”: Please tell us more about how this criterion was defined, could it be perhaps too arbitrary for the adolescent population in this context? How widely/not are phones/devices with internet access used to set the bar at: at least one medium once/+ a week?

Response:

The reviewers’ comment highlights additional opportunities to clarify the exposure variable. Additional sentences were added to clarify that “frequency of mass media exposure” was a categorical variable. Categorical levels were written and revised based off expertise around Nepal’s local context. The 5 categories were as follows: (0) never, (1) once or twice, (2) less than once a month, (3) once a month, (4) 2-3 times a month, and (5) every week. If adolescents answered yes to the most frequent exposure (5) across any of the mass media channels they were included in the “frequent exposure” group. 

The sentences were revised to reflect the clarification above as follows (line 223-226). Similarly, information on mass media exposure was collected by asking adolescents how often they watched TV, listened to the radio, read the newspaper, and used the internet--- never, once or twice, less than once a month, once a month, 2-3 times a month, or weekly. Frequent exposure to mass media was defined as having watched, listened, read, or used at least one medium once or more a week. 

Comment:

I appreciate the clear tables describing the results and the descriptions.

Table 4 – seems to be missing a word in the title

Response:

Table 4 title updated to “Table 4. Crude and multivariate associations of grandmothers’ residency and adolescent FP knowledge (n = 769)

Comment:

Discussion: I very much welcome the evidence provided in this section identifying relevant, recent projects and evidence to corroborate the role grandmothers play in SRH. But I would also like to invite the authors to re-engage with the ecological model to conceptually enhance their results. The authors present a helpful figure on the application of the model but the results also identify demographic patterns on grandmother co-residence and knowledge(s), so I suggest they could consider: how could Bronfenbrenner’s insights better inform these results beyond categorizing the variables in the individual, micro, exo and macro systems?

The limitations are clearly stated as the authors are working with an existing dataset. Their arguments for future research were aligned with my own reading of the study, calling towards exploring the knowledge exchange between co-resident grandmothers and adolescents: what types of knowledge are shared that could help better inform grandmothers’ role as advisors and influencers of FP and SRH more generally? A call that I also second and invite the authors to consider for their next research project.

Response:

The reviewer raises interesting questions that the authors agree are not fully addressed in the discussion. We acknowledge this as a limitation of the current study as it relies on the available survey design and data. See Lines 453 to 456. 

While this study uses systems analysis to identify the source of sexual and reproductive knowledge among the surveyed adolescents, the survey design and data do not support the analysis of the interactions and dependencies between individual and contextual systems. For example, interactions with peer groups and schools, as well as the adolescent’s sexual partnerships, were beyond the scope of this study. These nested systems and their synergetic effects, however, can play an important role in adolescent development, inviting or inhibiting adolescent FP knowledge. 

In our summary we agree that further work should leverage systems analysis to better understand knowledge exchange and interactions between grandmothers and adolescents. This understanding could support the design of more effective reproductive and sexual health programs. See lines 476-478.

---

## [Editor Report · Decision Letter 1]

28 Feb 2022

Beyond the mother-child dyad: Is co-residence with a grandmother associated with adolescent girls’ family planning knowledge?

PONE-D-21-35002R1

Dear Dr. Zevallos-Roberts,

We’re pleased to inform you that your manuscript has been judged scientifically suitable for publication and will be formally accepted for publication once it meets all outstanding technical requirements.

Kind regards,

Kannan Navaneetham, PhD

Academic Editor

PLOS ONE
---

## [Editor Report · Acceptance letter]

7 Mar 2022

PONE-D-21-35002R1 

Beyond the mother-child dyad: Is co-residence with a grandmother associated with adolescent girls’ family planning knowledge? 

Dear Dr. Zevallos-Roberts:

I'm pleased to inform you that your manuscript has been deemed suitable for publication in PLOS ONE. Congratulations! Your manuscript is now with our production department. 

Kind regards, 

on behalf of

Prof. Kannan Navaneetham 

Academic Editor

PLOS ONE